# An Efficient Analytical Approach to Visualize Text-Based Event Logs for Semiconductor Equipment

Gunwoo Lee [ID] and Jongpil Jeong *[ID]

Department of Smart Factory Convergence, Sungkyunkwan University, 2066 Seobu-ro, Jangan-gu, Suwon 16419, Korea; comlee@g.skku.edu
* Correspondence: jpjeong@skku.edu; Tel.: +82-10-9700-6284 or +82-31-299-4267

**Abstract:** Semiconductor equipment consists of a complex system in which numerous components are organically connected and controlled by many controllers. EventLog records all the information available during system processes. Because the EventLog records system runtime information so developers and engineers can understand system behavior and identify possible problems, it is essential for engineers to troubleshoot and maintain it. However, because the EventLog is text-based, complex to view, and stores a large quantity of information, the file size is very large. For long processes, the log file comprises several files, and engineers must look through many files, which makes it difficult to find the cause of the problem and therefore, a long time is required for the analysis. In addition, if the file size of the EventLog becomes large, the EventLog cannot be saved for a prolonged period because it uses a large amount of hard disk space on the CTC computer. In this paper, we propose a method to reduce the size of existing text-based log files. Our proposed method saves and visualizes text-based EventLogs in DB, making it easier to approach problems than the existing text-based analysis. We will confirm the possibility and propose a method that makes it easier for engineers to analyze log files.

**Keywords:** log management; log parsing; equipment; EventLog; timescaledb; PostgreSQL

## 1. Introduction

Many semiconductor equipment manufacturers store everything that occurs during the operation of the equipment in a log [1,2]. The log contains valuable information that helps understand the system execution status and accurately identifies system errors. In addition, engineers monitor system and equipment debugging for abnormal operations and errors and include process progress information for future debugging and analysis [3]. The Log stores all the information generated from various modules (e.g., Process Module, Transfer Module, Individual Components) that compose the equipment.

Logs are not standardized, and some equipment manage different log levels depending on the equipment manufacturer. In general, there are the EventLog, the Process Log, the Alarm Log, the Lot History Log, and the Debug Log. The EventLog stores everything according to the passage of time and is organized as a time series. The EventLog is text-based, so it is hard to see. In the case of a long process, it is composed of several files, and it takes a long time to analyze because it is necessary to understand the relationship between many files [4,5]. The Process Log records information of the parameters related to recipes while the process is in progress. The items included in the Process Log are generally provided with functions to be defined by the user, and the Process Log is created based on the defined items. The Alarm Log consists of Error, Warning, and Information and stores the contents of all alarms that have occurred in the equipment. In terms of troubleshooting, when an alarm occurs, the equipment can clear several alarms while the engineer is taking action. If the user selects the wrong options, the alarm remains unresolved, and it has become a common practice to capture various user interactions in the EventLog and store them in a log file for later analysis [6,7]. The Lot History Log contains the history information of

the lot in the equipment. Users can check information about the progress history through this log. The Debug Log contains detailed information for troubleshooting when there is a problem with the device. For example, there are Code Level Stack Dump and Snapshot logs. In general, there is a System Debug Log for debugging when a problem occurs in the system, and the Debug Log at the module level is used for each module as needed. Various types of logs are suitable for various purposes, and the contents and types of logs can vary significantly from system to system, even between components within the system. The Event Log is the most common. It stores all contents that occur while all equipment companies operate equipment. The Event Log is usually the first log that engineers check to determine the cause of a problem in the equipment. The Event Log has a vast amount of logging as all the history of the equipment has been stored, and it is difficult to grasp the exact causal relationship because events that have occurred in multiple modules are recorded in one log.

Additionally, if the problem occurred in the past, it is often difficult to analyze. In some cases, analyzing a problem device and several devices that do not have a specific time may exceed human analysis capabilities. In terms of log file size, when saving text-based logs, it is necessary to minimize the file size and efficiently save and manage it. Despite the enormous value hidden in the log files, how to analyze them effectively remains a big challenge [8].

This study started with the idea that engineers had difficulties with text log analysis and visualizing the text-based EventLog would help engineers analyze it. Customers are requesting more sensor data at a faster cycle. The saved text-based EventLog is compressed and stored on the hard disk. In particular, the disk size of computers that have been used in the past is not large. So, many types of equipment are using the minimum period of keeping it. If an EvnetLog file becomes oversized, this reduces the period that it can be kept. It is often difficult to check the EventLog because it has already been deleted after a certain period. If the EventLog size is large, many files cannot be stored, so a method to reduce the EventLog is needed. We were looking for a way to reduce the size even by changing the file structure of the existing EventLog. The file structure we found is Technical Data Management Streaming (TDMS).

To check the usefulness of TDMS, file sizes were compared, and actual memory usage was also compared. In conclusion, TDMS was able to store more information under the same conditions. TDMS can be one alternative to reduce the file size. Many EventLogs need an analytics system to experiment with in different ways, and we designed and implemented an EventLog analysis system to analyze, targeting the EventLog. This paper presents an example that various expressions are possible by extracting and visualizing meaningful information from the EventLog of the existing Text Base through this system.

The paper is organized as follows. Section 2 describes the background and related research. Section 3 describes the configuration of the log analysis system and the DB design. Section 4 discusses the experimental results, and finally, Section 5 presents the conclusions.

## 2. Background and Related Research

Many logs made in semiconductor equipment must be managed well. The hard disk of the equipment controller has a limited capacity and cannot store files indefinitely. Therefore, it is essential to automatically compress [9] past logs and delete them after a certain period. If the EventLog is deleted due to insufficient space on its hard disk, the EventLog needs to be saved for as long as possible because there is no way to verify the problem. To do so, we needed a way to reduce the size of Log. By default, log-based text is compressed and archived. However, we needed to change the same information to a different format and reduce the size. Previous studies have also proposed a Comprehensive Log Compression(CLC) method that integrates data compression online [10] or uses frequent patterns and compression representations to identify repetitive information in large log files generated by communication networks [11]. However, these studies have reduced file size through compression without changing the structure of existing data. Parsing must be used

to obtain the information it wants from the Log file [12,13]. Depending on implementing parsed information, we also use the method of finding anomaly identification [14–16]. As an example of log analysis, the FIU aims to automate the process model [17] and facilitate data analysis on the system event log by combining the equipment status and event text analysis based on log files for the purpose of equipment productivity and diagnostic evaluation. There are studies on the design and implementation of an integrated system called Log Analysis Platform (FLAP) [18], and several event clustering proposal studies on log files in a network environment [19,20].

More diverse studies have been conducted from the process mining perspective. In particular, studies have been conducted on algorithms and various models. Typical algorithm examples include pattern mining studies using logCluster algorithms [21], algorithms for detecting if logs are complete [22], process-oriented data mining algorithms and model configurations that provide insight into organizational processes using event log data [23], and algorithm studies for visualizing large-scale parallel process models [24–27]. In addition, a study on a general approach to mining the personal information protection process using the information accessed to the system log [28], a study on personal information protection [29], event abstraction [30], knowledge extraction [31], operation Research [32], security vulnerabilities [33], quality-aware semi-automatic approach to extracting EventLogs from relational data [34], and event log prediction studies using RNN for utilization of EventLogs were performed [35].

From a mining perspective, some of the studies on information extraction and visualization are similar to ours. However, the visualization target is the visualization part of the model and differs from ours as the Logfile contents are not visualized in various ways. In order to store more information in the field of semiconductor equipment, semiconductor equipment is also offered as an optional PC for separate data storage. Data-Distribution Service for Real-Time Systems (DDS) for high-speed data logging, such as Process Log, is used to store many data in Log, but not all semiconductor equipment uses DDS. To evaluate DDS communication performance based on models built within the IEC 61,499 standard and compare it with existing socket-based solutions [36,37] and evaluate equipment productivity and diagnostic purposes, automating model composition and processes based on log files [38] were investigated.

DDS is focused on storing more data. Furthermore, research based on log files is similar in part to a parsing Log but different from the research we are conducting because it focuses on automating processes. We have looked into many previous studies but have not found any similar examples of research to the present study. Our research aims to study a new Format for more extended storage of EventLogs generated by equipment and make it possible for engineers to identify problems by visualizing text-based EventLogs quickly. An EventLog is typically stored in ASCII format as Text. File Format is also an essential factor if storage needs to be stored and frequently updated. When logging data in 0.1 s, the system has extensive logging, which can also affect communication. Therefore, it is essential to select the Data Format that can produce the best performance when compressing files in the EventLog.

National Instrument developed technical data management streaming (TDMS) to store measurement data. A three-step structure consisting of Root, Group, and Channel can place another group under one Root and yield the resulting channel. Above all, unlimited user properties can be added to each step, making it suitable for storing large amounts of sensor data, such as semiconductor equipment. The aim of this study is to verify whether TDMS can reduce and utilize the EventLog's file size in the future by comparing the existing ASCII saved EventLog with the TDMS-converted file. In addition, by visualizing the EventLog, we suggest what contents can be displayed compared to the text-based one.

## 3. Log Analysis System Configuration and DB Design

EventLog files of semiconductor equipment were collected and studied. The overall system development language used R, and Database configured the system using timescaledb in PostgreSQL.

### 3.1. Configuration of EventLogs

EventLogs show what happened in the system based on the source of the event. The history of events is displayed chronologically, making it easy to see what happened from the past to the present. The event log consists of the following:

- Record the contents of all user actions in the UI. For example, the operator starts or stops Lot with the manual and records everything he or she looks upon a particular screen.
- Records the events that occurred for all modules. For example, if you open and close the valve of an installation or run a command to raise the temperature to a certain temperature, you will record the information in the EventLog that increases the temperature.
- In the EventLog, information about failures in the event of a facility failure and in the event of a failure resolution are also stored.
- Records all information related to scheduler behavior.
- Displays information about each module operating situation.

The explanation of each item of the content shown as an example in Figure 1 is as follows.

- Log Time: The Log Time indicated the time stamp when the CTC server receives the event and records the logging on the log file.
- Issue Time: The issue time indicates the time stamp when the issuer reports the events.
- Event ID: Each event has a specific eventID and this ID section shows a number of it.
- Name: It is an identifier of the parent category of each event.
- Issuer: Each event is published by each issuer, and the issuer can be either device level or image level.
- Event Text: Event Text provides more detailed information about which event has occurred.

| Log Time | Issue Time | ID | Name | Issuer | Event Text |
|---|---|---|---|---|---|
| 21/06/16 10:44:16 | 21/06/16 10:44:14 | 1111300104 | MoveStart | PM1GasVacuumSystemGas4OutletValve_Gas4 | Started move to position closed. |
| 21/06/16 10:44:16 | 21/06/16 10:44:14 | 1111300103 | MoveEnd | PM1GasVacuumSystemGas4OutletValve_Gas4 | Completed move to position closed taking 0 ms. |
| 21/06/16 10:44:16 | 21/06/16 10:44:14 | 1111300136 | MoveStart | PM1GasVacuumSystemManifold_AManifoldADivertValve | Started move to position closed. |
| 21/06/16 10:44:16 | 21/06/16 10:44:14 | 1111300135 | MoveEnd | PM1GasVacuumSystemManifold_AManifoldADivertValve | Completed move to position closed taking 0 ms. |
| 21/06/16 10:44:16 | 21/06/16 10:44:14 | 1111300138 | MoveStart | PM1GasVacuumSystemManifold_AManifoldAOutletValve | Started move to position closed. |
| 21/06/16 10:44:16 | 21/06/16 10:44:14 | 1111300137 | MoveEnd | PM1GasVacuumSystemManifold_AManifoldAOutletValve | Completed move to position closed taking 0 ms. |
| 21/06/16 10:44:16 | 21/06/16 10:44:14 | 1111300140 | MoveStart | PM1GasVacuumSystemManifold_AManifoldAPurgeGas | Started move to position closed. |
| 21/06/16 10:44:16 | 21/06/16 10:44:14 | 1111300139 | MoveEnd | PM1GasVacuumSystemManifold_AManifoldAPurgeGas | Completed move to position closed taking 0 ms. |
| 21/06/16 10:44:16 | 21/06/16 10:44:14 | 1111300164 | MoveStart | PM1GasVacuumSystemManifold_BManifoldBOutletValve | Started move to position closed. |
| 21/06/16 10:44:16 | 21/06/16 10:44:14 | 1111300163 | MoveEnd | PM1GasVacuumSystemManifold_BManifoldBOutletValve | Completed move to position closed taking 1 ms. |
| 21/06/16 10:44:16 | 21/06/16 10:44:14 | 1111300166 | MoveStart | PM1GasVacuumSystemManifold_BManifoldBPurgeGas | Started move to position closed. |
| 21/06/16 10:44:16 | 21/06/16 10:44:14 | 1111300165 | MoveEnd | PM1GasVacuumSystemManifold_BManifoldBPurgeGas | Completed move to position closed taking 1 ms. |
| 21/06/16 10:44:16 | 21/06/16 10:44:14 | 1111306109 | MoveStart | PM1GasVacuumSystemManifold_CManifoldBOutletValve | Started move to position closed. |
| 21/06/16 10:44:16 | 21/06/16 10:44:14 | 1111306108 | MoveEnd | PM1GasVacuumSystemManifold_CManifoldBOutletValve | Completed move to position closed taking 0 ms. |
| 21/06/16 10:44:16 | 21/06/16 10:44:14 | 1111301373 | MoveStart | PM1GasVacuumSystemManifold_CManifoldCOutletValve | Started move to position closed. |
| 21/06/16 10:44:16 | 21/06/16 10:44:14 | 1111301372 | MoveEnd | PM1GasVacuumSystemManifold_CManifoldCOutletValve | Completed move to position closed taking 0 ms. |
| 21/06/16 10:44:16 | 21/06/16 10:44:14 | 1111301017 | MoveStart | PM1GasVacuumSystemRoughValve | Started move to position closed. |
| 21/06/16 10:44:16 | 21/06/16 10:44:14 | 1111301016 | MoveEnd | PM1GasVacuumSystemRoughValve | Completed move to position closed taking 0 ms. |
| 21/06/16 10:44:16 | 21/06/16 10:44:14 | 1111301324 | MoveStart | PM1GasVacuumSystemGasOInletValve_GasO | Started move to position closed. |
| 21/06/16 10:44:16 | 21/06/16 10:44:14 | 1111301323 | MoveEnd | PM1GasVacuumSystemGasOInletValve_GasO | Completed move to position closed taking 1 ms. |

**Figure 1.** Example of an EventLog.

### 3.2. Architecture Design

An example of an EventLog is the overall system configuration, as shown in Figure 2, divided into two parts. The first experiment proceeded as follows. EventLogs stored based on Text were collected in the equipment. We developed a TDMS Parser that can store the same information and collected EventLogs stored in TDMS format. We compared the two groups of File Size and measured the loading rate through the program that can view EventLogs. The results of the experiment are explained in Section 4.

The second development part was carried out as follows. The EventLog was collected by period and model to understand the overall structure. The EventText described in Figure 1 contained various information, which we extracted and provide in a schematic. Text information was drawn in various forms and some contents that were considered meaningful were identified. Based on this information, we created a table in DB and saved the information of the EventLog in a table that meets the query criteria. We developed the program using the R program information stored in each table.

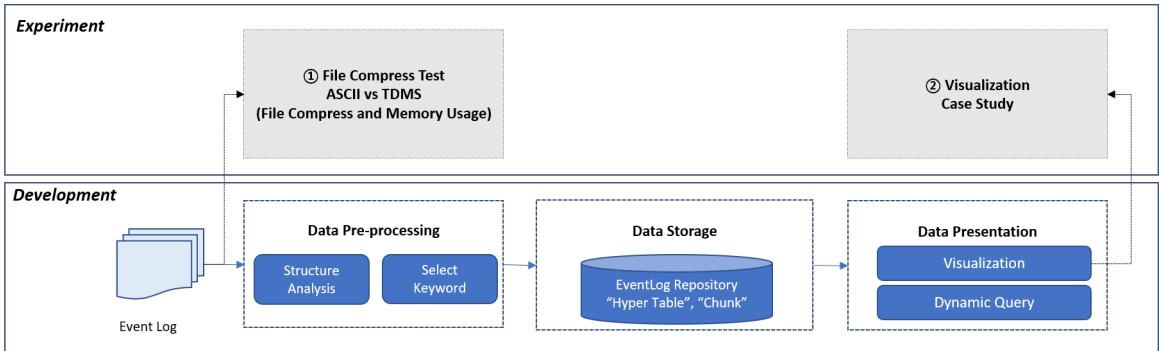

**Figure 2.** System Architecture.

### 3.3. Configuration of PostgreSQL and Timescledb

To select a database, we tested various types of DBs. The EventLog is time-series data that stores data over time. Sometimes it is necessary to delete the old data set from the DB. Deletion may be complicated because the table structure is not Hypertable or Chunk, so that it can be easily separated from data integrity. Timescaledb, an extension of PostgreSQL, provides time-series database properties, and EventLog DB Tables can be designed as Hypertables. The EventLog data table, with time and partition, is implemented as a Hypertable, and the EventLog is collected and saved in Figure 3 Hypertable and Figure 4 Chunk. In addition, when the chunk table deletion policy is registered, it provides a read-only data compression function that saves space by up to 90%, and provides automatic cluster command execution for files that are no longer changed. If it uses this function, it can manage the DB efficiently.

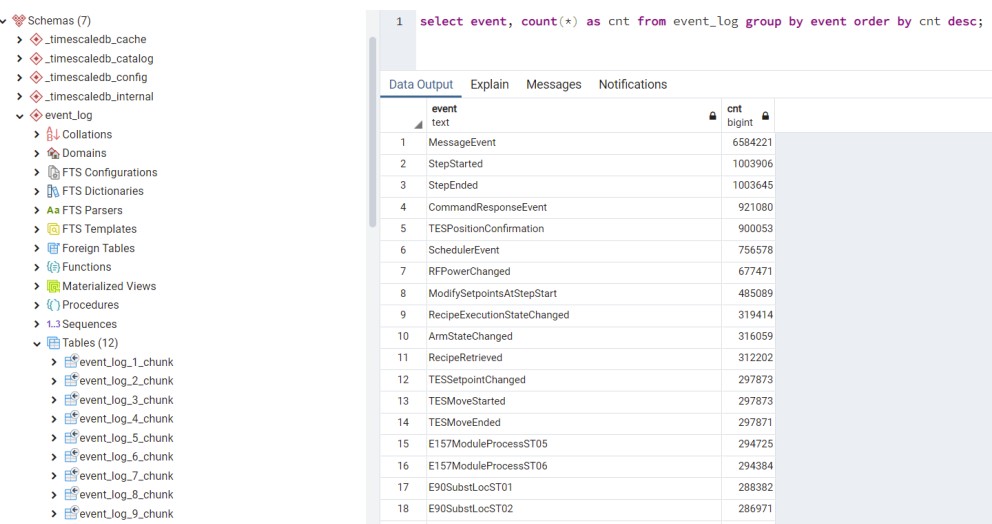

**Figure 3.** Hypertable (Logical database table).

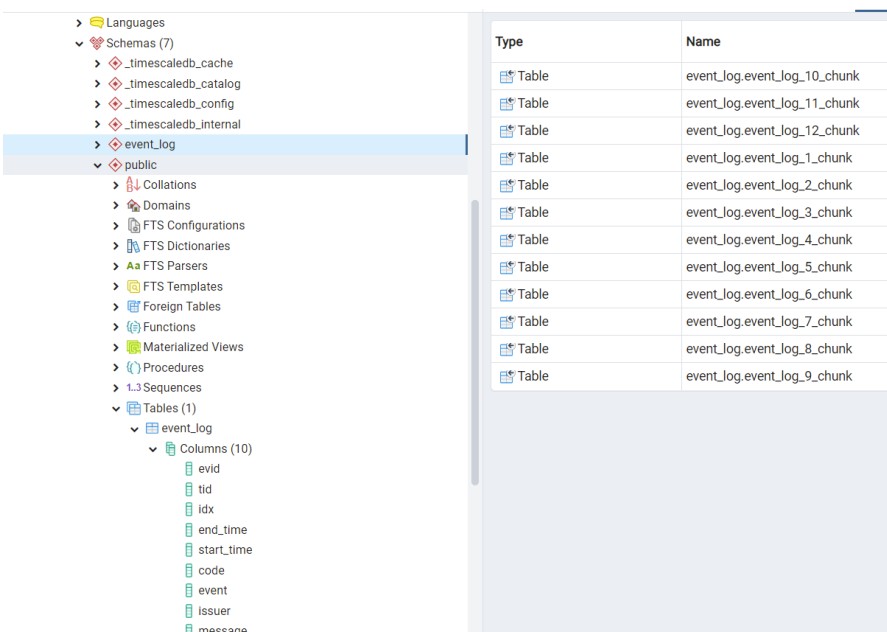

**Figure 4.** Chunk (Physical table).

## 4. Experiment and Results

### 4.1. Experiment Environment

EventLogs were collected for one month for the Etch equipment in the Semiconductor Lab. The instrument generated two EventLogs every hour. In the EventLog, more than 1000 issuers were recording data. Among the many EventLogs, we have experimented with the EventLog, that has the most extensive number of issuers. The experiment was conducted in the following manner.

- Save the EventLog file in DB.
- Using R Code, we figured out data properties while drawing various pictures in the DB.
- The experiment was carried out by grouping into several groups.
- Checked the data of several files in a row.

Through this process, we identified the types of meaningful data. Therefore, this Prototype Test is essential. Finally, we made a DB table after confirming the data on the screen that we show to the user.

### 4.2. ASCII vs. TDMS Comparison Result

The EventLog is structured based on Text. Additionally, the contents saved in the log are determined by the log level designated by the user. For example, if the log level is 1, only the most basic details are saved in the log. In the highest Level 5, all fine details are saved in the log, and the log file size increases. It is not easy to store large amounts of information quickly. Further, the use of disk may be restricted due to the increase in the file size of the EventLog. Therefore, we checked if TDMS is an alternative to reducing the size when saving Text Log. First of all, we checked the type of TDMS that is configured.

TDMS can be expressed in a total of three file structures, as shown in Figure 5. A file can be created as a Physical File Format, configured as a Logical File Format, or configured as a Perceive File Format. An EventLog consists of Time-Series, so it is usually organized in the form of a Physical File Type. However, the contents inside can be extended to either the logical type or the Perceive file type. We analyzed the format of the existing EventLog and configured Objects and Properties according to the contents. A function was made to convert to TDMS. We evaluated the compression rate and memory usage rate with the existing text-based file and the new TDMS file format.

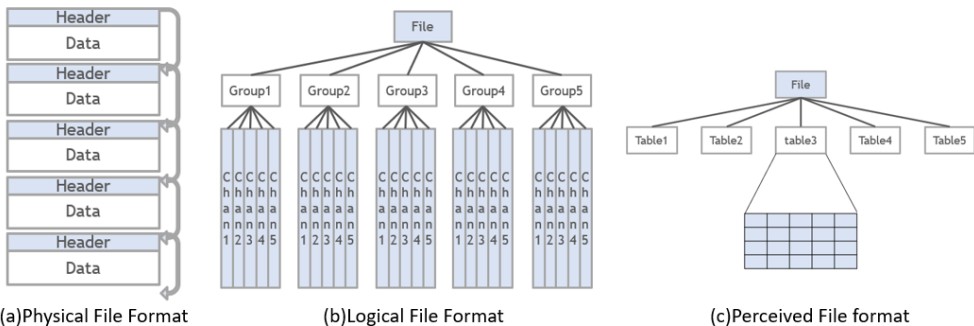

**Figure 5.** Form of TDMS file.

Experimental results in Figure 6, the ASCII-based EventLog increased in Loading time in direct proportion to the length of the file increase. On the other hand, in TDMS, the change in speed was insignificant even when the file size increased. This is because when storing the same amount of data, TDMS is structured compared to Ascii, and the size can be reduced. For example, in ASCII-based LogFile, time can be displayed in the year, month, day, hour, minute, second, and millisecond. In the text-based EventLog, time is configured in ASCII. However, TDMS can convert to Double. Not only time but also many numerical data stored in the EventLog can be reduced in size when converting to TDMS. Therefore, TDMS can reduce the file size and is faster compared to ASCII.

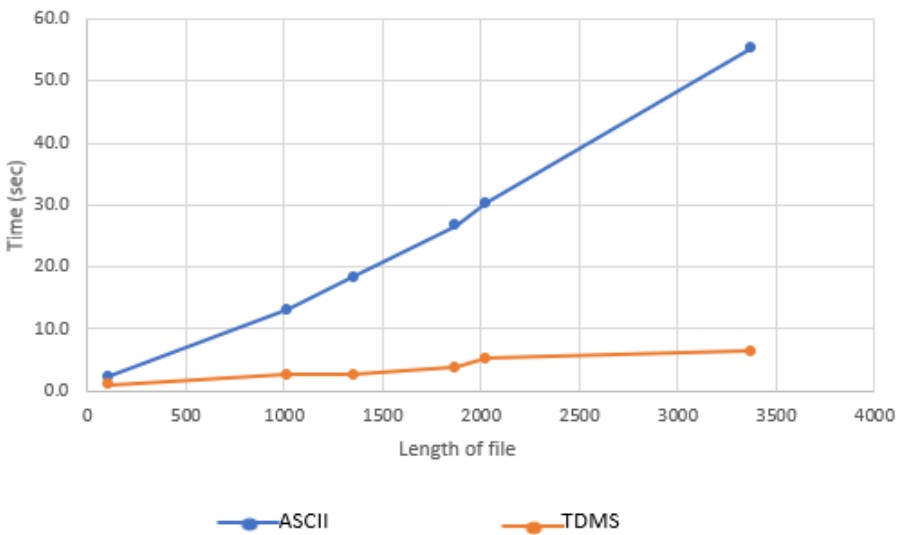

**Figure 6.** ASCII vs. TDMS Loading Speed Comparison.

As a result of comparing memory use under the same conditions, it was found that the longer the file length, the higher the memory use of an ASCII-based EventLog. The constant memory usage of TDMS was maintained as shown in Figure 7.

In conclusion, as shown in Table 1, TDMS files are smaller in size, more compressed, and use less memory than ASCII files. A 22.5 min recipe was run on the equipment, and data were collected from 1582 channels at 0.1 s intervals. The collected files were collected in two types, a general ASCII file, and a TDMS file. The file size was reduced to 51% for uncompressed files compared to ASCII by TDMS and 64% for compressed files. In addition, the compression ratio of TDMS was 1% higher than that of ASCII.

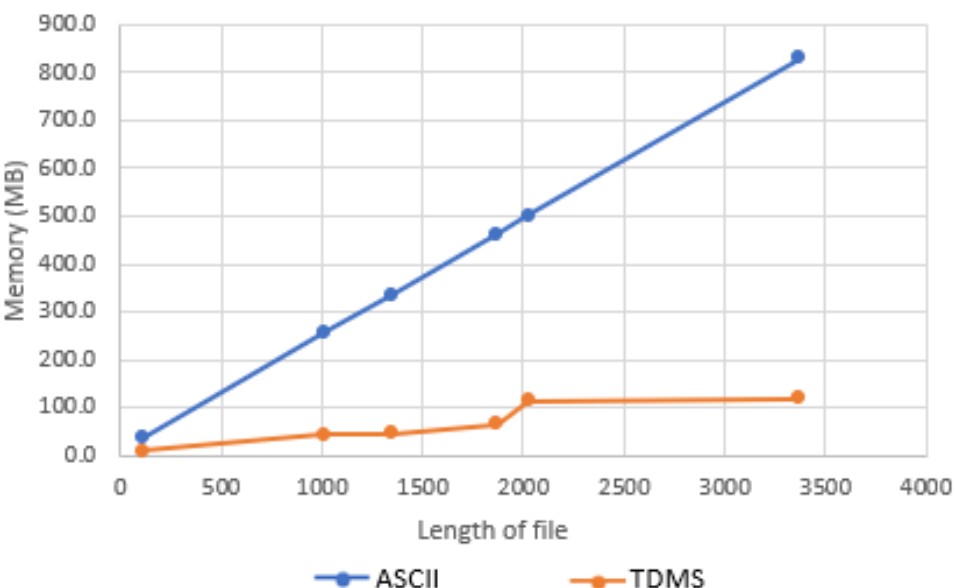

**Figure 7.** ASCII vs. TDMS Memory Usage Comparison.

**Table 1.** ASCII vs. TDMS Compression Ratio Comparison.

| 1582 Channels (22.5 Min), 10 Hz | Unzipped [MB] | Zipped [MB] | Compression Rate |
|:---:|:---:|:---:|:---:|
| ASCII | 201.8 | 10.0 | 95% |
| TDMS | 99.9 | 3.5 | 96% |
| Size Reduction | 51% | 64% | |

*4.3. EventLog Visualization Results*

After configuring the system and storing the EventLogs in the DB, the result of the analysis was as follows. Existing text-based EventLogs are not intuitive, and it often takes a long time to find the exact location. By expressing the EventLogs in various forms, we can more easily find the cause of the problem and understand the overall contents.

- Expressing the Frequency of Event Occurrence by Module:
  As shown in Figure 8, The subject generating the event is recorded in the EventLog, and it was designated as an issuer. If it draws the cumulative chart for each event issuer, it can yield the following screen. This type can be used in various ways. For example, you can view alarms by type in the Alarm Log, and you can also check the total number of alarms.
- Expressing the Amount of Change Over Time:
  There are cases where one needs to monitor how parameters change over time as shown in Figure 9. For example, it is crucial to managing the accumulated Radio Frequency (RF) On-time in the process chamber in Etch equipment. In this case, it is possible to draw a trend chart as follows by overlapping several EventLogs and displaying several parameters together.
- Monitoring Success and Failure by Period:
  A macro is a kind of script that automatically checks things that have been manually checked by the operator in the equipment. Some of these macros are executed daily, and there are cases where macros are executed under certain conditions. We can check whether the executed macro was executed typically or was abnormally terminated. Pass or Fail information may be utilized as a future monitoring feature as in Figure 10. It can be used to monitor the entire equipment and to monitor the equipment that frequently fails.
- WaferMap; Map information about wafers is stored in the EventLog, and a WaferMap can also be drawn using this information. WaferMap information is vital information.

With this information, if it can draw a map before and after the process proceeds, it can judge the process result through the wafer map as in Figure 11. These Map data can also be used to compare before and after Etch.

- Check the Idle Section: In semiconductor equipment, the event when the recipe starts and ends is recorded. If we express this below, we can check where the chamber is idle. If we mark the start and end parts at the beginning and end of the recipe and expand to information of recipe parameters for each step, we can analyze how the equipment operated at that time. In addition, it is possible to compare why the delay between two or more pieces of equipment occurs by using this information. The equipment performance degradation can be analyzed by checking and analyzing the delayed section until the recipe starts and ends. It can also compare the data between two equipment with differences in throughput to see the delay interval. Through visualization, it can easily find the section where Idle occurs as in Figure 12.

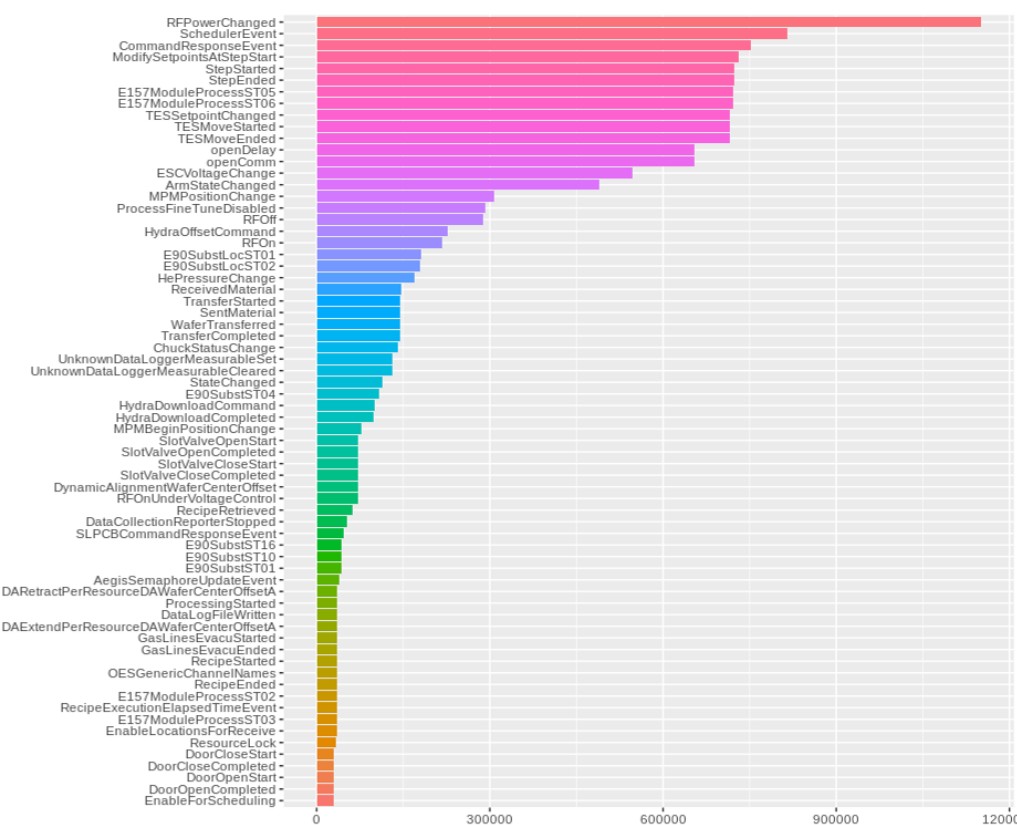

**Figure 8.** Event Occurrence Frequency by Module.

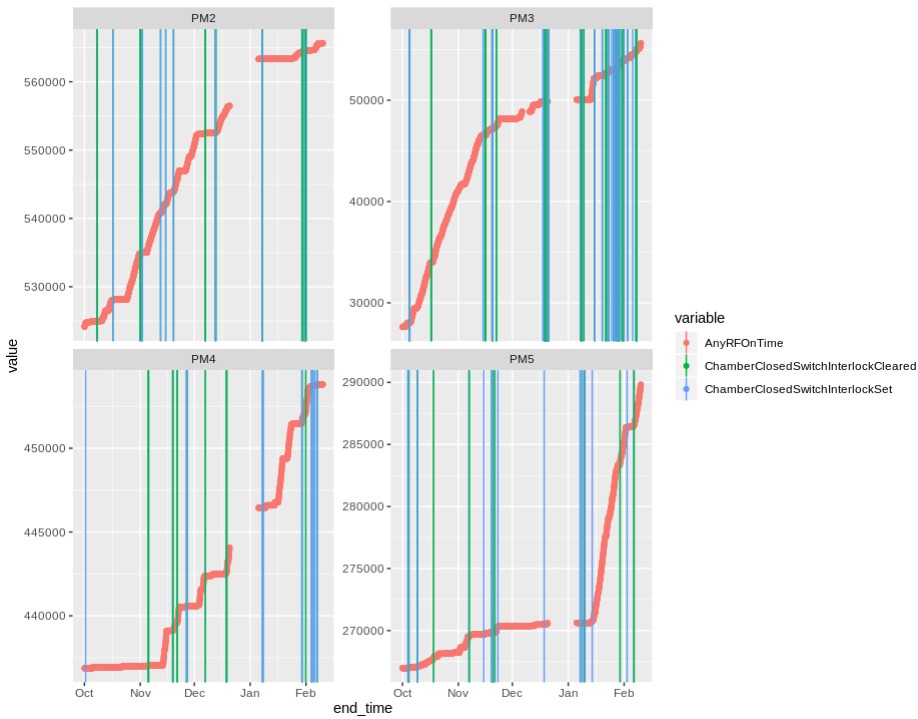

**Figure 9.** Trend Chart of Multiple Parameters for The Amount of Time Change.

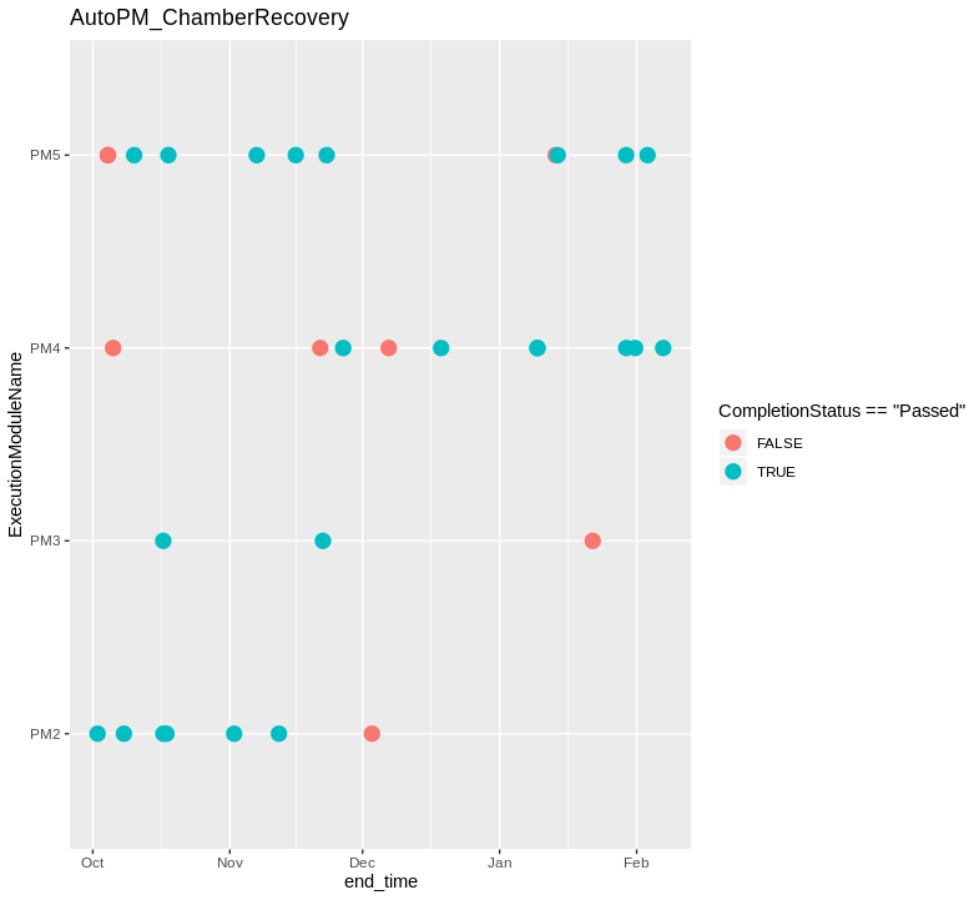

**Figure 10.** Success and Failure Monitoring.

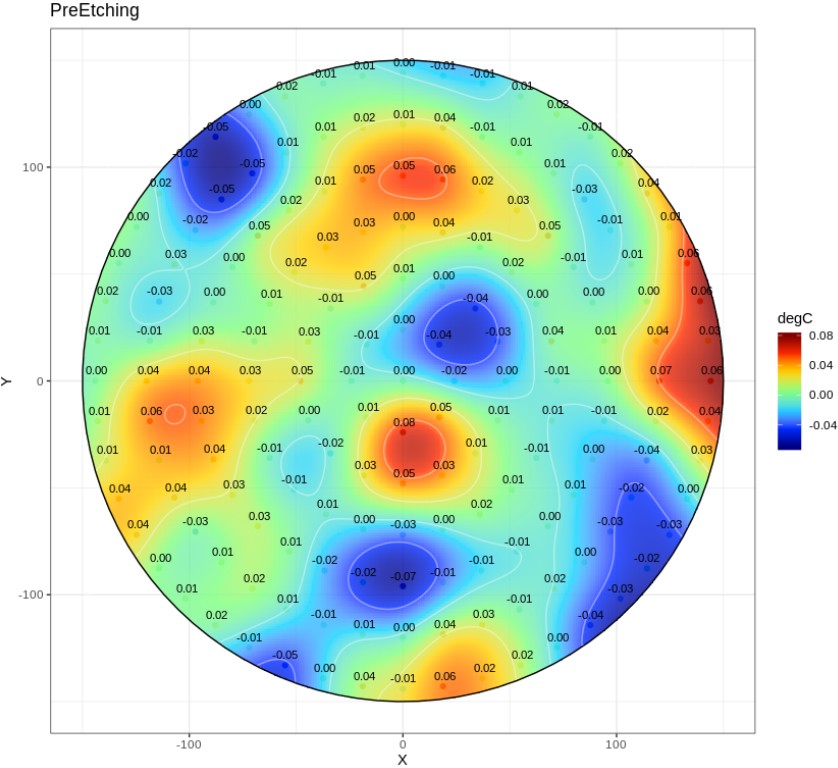

**Figure 11.** WaferMap .

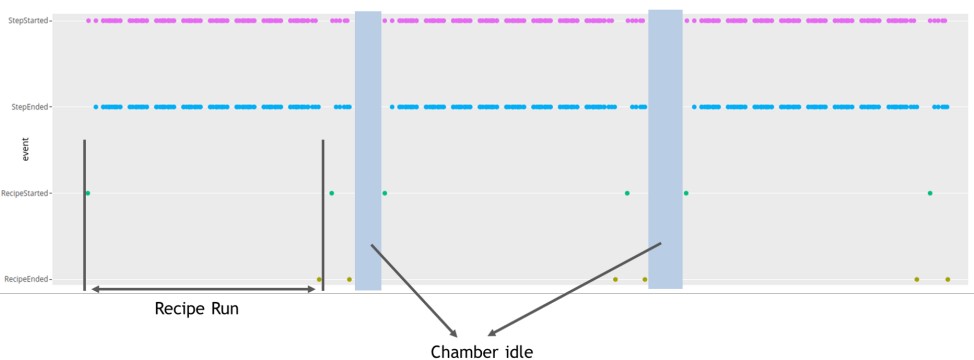

**Figure 12.** Check the Idle Section.

## 5. Conclusions

In this study, two experiments were conducted with the EventLog. The file size of the EventLog was reduced to 51% for uncompressed files compared to ASCII by TDMS and 64% for compressed files. In addition, the compression ratio of TDMS was 1% higher than that of ASCII. Regarding the EventLog analysis of the equipment, the new value of utilizing the text-based EventLog was confirmed through experiments on various cases that can be performed with the event log through the system. In particular, when analyzing multiple files simultaneously, it was visually displayed when this system was used, and it was more convenient than analyzing with existing text for engineer analysis. From the perspective of big data, log files are another area of research. Many companies and researchers are studying the EventLog and the System Log of computer systems, but the field of log research of semiconductor equipment is not actively studied. Analyzing various log information, creating new value from log files, and expanding it will be used in conjunction with a monitoring system and a big data system. Through LogFile analysis, we learned that many analyzes could be done using Log. The purpose of the final study we are going

to carry out is not just show a single EventLog. Using this system, we want to store various logs in the semiconductor equipment in the DB and compare between equipment. By using the data stored in the system, we can check the reason for the difference in the performance of the equipment and check when this problem started. For this reason, log file analysis is an essential factor in the big-data analysis of semiconductor equipment. Considering when hundreds of pieces of equipment are connected, we plan to conduct further research on the pre-treatment. In addition, the Equipment Data Acquisition (EDA) Client, which is used as a standard semiconductor protocol, collects all the data of semiconductor FAB in real time. We are very interested in this data, and we plan to conduct further research on Log Parsing in the EDA Client.

**Author Contributions:** G.L.; conceptualization, G.L.; software, J.J.; validation, G.L.; writing—original draft preparation, J.J.; writing—review and editing, G.L.; visualization, J.J.; supervision, J.J.; project administration. All authors have read and agreed to the published version of the manuscript.

**Funding:** This research was supported by the MSIT (Ministry of Science and ICT), Korea, under the ITRC (Information Technology Research Center) support program (IITP-2021-2018-0-01417) supervised by the IITP (Institute for Information & Communications Technology Planning & Evaluation) and the Smart Factory Technological R&D Program S2727186 funded by Ministry of SMEs and Startups(MSS, Korea).

**Institutional Review Board Statement:** Not applicable.

**Informed Consent Statement:** Not applicable.

**Data Availability Statement:** Not applicable.

**Acknowledgments:** This work was supported by the Smart Factory Technological R&D Program S2727115 funded by Ministry of SMEs and Startups (MSS, Korea).

**Conflicts of Interest:** The authors declare no conflict of interest.

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
