# Peer review of "An Efficient Analytical Approach to Visualize Text-Based Event Logs for Semiconductor Equipment"

_applsci, doi:10.3390/app11135944_

Round 1

Reviewer 1 Report

I am fairly certain that hidden this paper are some good ideas concerning semiconductor logs.  But there is a major ambiguity in the presentation.  Are the authors (1) proposing a normal form for semiconductor logs because "Logs are not standardized," (2) proposing a novel file organization for such logs or (3)  is the proposed contribution the addition of novel methods of access and visualization.  The paper at different junctures seems to be making one or all three of these claims without ever being quite explicit.  I have read it several time without success to discern its main purpose.

I have noted that the paper contains many non- idiomatic forms of expression that interfere with comprehension.

Author Response

Dear Reviewer,

Thank you for reviewing the resubmitted paper. We have reviewed each of the items that you mentioned, and we will deliver our opinion and the revised menu script document.

We sincerely appreciate your efforts to review the journal and look forward to hearing from you soon.

Sincerely yours,

GunWoo Lee, Graduate Student.

Department of Smart Factory Convergence (Advisor: Jongpil Jeong)

College of Software

Sungkyunkwan University (SKKU)

2066 Seobu-ro Jangan-Gu, Suwon 440-746, KOREA (South)

Tel: +82-31-299-4260(Office)/ +82-10-4932-1802(Mobile)

Reviewer 2 Report

Log analysis is nowadays a very significant task since, once we are able to manage with the possibly large amount of collected reports, interesting knowledge can be extracted from logs, allowing to know more about the studied process and how it is going. Authors run an analysis over some samples of event logs in an experiment, describing the obtained results. While the proposal and the original idea can be interesting, the process is poorly described, the text is confusing and the results are misleading and unclear.

The state of the art section (Section 2) is hard to follow. Since it establishes the background and previous approaches that justify the aim of the paper, it must be clearly stated what are the authors' intentions, and how they follow the existing research.

Section 3 needs to explain in more detail the proposed architecture. How are the log files structured? What can we expect from the analysis of those log files? Is it really significative the obtained differences between ASCII and TDMS logs, compressed or uncompressed?

Experiment design needs to be improved and clarified. Which tools have authors used? What can we conclude from the presented graphs?

Log analysis problem is a challenging one, but the paper needs to improve its structure and discussion in order to satisfy its proposal.

Author Response

(The authors gave the same response as above.)

Reviewer 3 Report

Paper as a lot of typos, specially at the bottom of the second page.

Figure 4 and Figure 5 had some typo at the X axis text; file light are not measured in Secs.

Figure 6, 7, 8 and 9 illustrate several applications of the logged data, but that not clear why but it is not noticeable why they are restricted from the TDMS format. Couldn't we have it with a munin (https://munin-monitoring.org) like tool?

Reading the paper is not clear what do you mean with EventLog. I guess some reference would help a lot.

Author Response

(The authors gave the same response as above.)

Reviewer 4 Report

This manuscript is an article about the visualization of text-based event logs for semiconductor equipment and this work is of interest to the scientific community and IC manufacturing engineers. This article is adequate for a dynamic journal of sustained academic excellence like Applied Sciences, but the quality of the manuscript must be strengthened to be accepted.

Many aspects of the manuscript must be improved. I strongly advise the authors to rewrite all the Section 2 (this is not a formal presentation of the background and related work). Also, please give more details for your architecture and present a database model except for the print screens. The results section is good but I believe that you must improve your descriptions and finally, the conclusions should present in more detail the problems in the industry, the highlights of this research results, and future implementations using big data and AI.

Author Response

(The authors gave the same response as above.)

Round 2

Reviewer 2 Report

Authors address the challenge of analyzing large amounts of information as it can be found in event logs, for many problems and processes in the real world. Actually, they focus on the problem of semiconductors equipment manufacturing.

The paper compares the analysis of raw texts in ASCII formats, compared to analysis of structured logs, by means of tools as Technical Data Management Streaming. Storing log information in a database results in more valuable and understandable results.

Authors have noticeable improved the original manuscript, correcting the English grammar, and focusing in some of the flaws addressed by the reviewers. I have to acknowledge the authors for their effort.

There are some minor typos in the text. For example, in page 2, "Evnetlog".

Author Response

Dear Reviewer2,

Thank you for reviewing the resubmitted paper. We have reviewed each of the items that you mentioned, and we will deliver our opinion and the revised menu script document.

We sincerely appreciate your efforts to review the journal and look forward to hearing from you soon.

Sincerely yours,

GunWoo Lee, Graduate Student.

Department of Smart Factory Convergence (Advisor: Jongpil Jeong)

College of Software

Sungkyunkwan University (SKKU)

2066 Seobu-ro Jangan-Gu, Suwon 440-746, KOREA (South)

Tel: +82-31-299-4260(Office)/ +82-10-4932-1802(Mobile)

Reviewer 3 Report

Most of the issues were completely addressed by you, but I am not happy yet with Eventlog answer. I mean, what is Eventlog: is it the log files the semiconductor equipment produces during its operation? Does it has any standardized format? Are you able to reference it? 

Author Response

Dear Reviewer3,

Thank you for reviewing the resubmitted paper. We have reviewed each of the items that you mentioned, and we will deliver our opinion and the revised menu script document.

We sincerely appreciate your efforts to review the journal and look forward to hearing from you soon.

Sincerely yours,

GunWoo Lee, Graduate Student.

Department of Smart Factory Convergence (Advisor: Jongpil Jeong)

College of Software

Sungkyunkwan University (SKKU)

2066 Seobu-ro Jangan-Gu, Suwon 440-746, KOREA (South)

Tel: +82-31-299-4260(Office)/ +82-10-4932-1802(Mobile)

Reviewer 4 Report

Firstly, I would like to mention that the authors improved their text based on my comments.

In Intro and Related work sections please split your text into paragraphs to improve the clarity of your work.

Please, take care of the bullets on page 4, use bold or ": "to make a distinction between the items and the explanatory text.

Generally, please give a strong effort to improve your English writing style, and write in a formal style the Abstract.

Author Response

Dear Reviewer4,

Thank you for reviewing the resubmitted paper. We have reviewed each of the items that you mentioned, and we will deliver our opinion and the revised menu script document.

We sincerely appreciate your efforts to review the journal and look forward to hearing from you soon.

Sincerely yours,

GunWoo Lee, Graduate Student.

Department of Smart Factory Convergence (Advisor: Jongpil Jeong)

College of Software

Sungkyunkwan University (SKKU)

2066 Seobu-ro Jangan-Gu, Suwon 440-746, KOREA (South)

Tel: +82-31-299-4260(Office)/ +82-10-4932-1802(Mobile)
